# Primary Care Patient Interest in Multi-Cancer Early Detection for Cancer Screening

**DOI:** 10.3390/jpm13111613

**Published:** 2023-11-16

**Authors:** Ronald E. Myers, Mie H. Hallman, Ayako Shimada, Melissa DiCarlo, Kaitlyn Davis, William T. Leach, Hattie Jackson, Amanda Indictor, Christopher V. Chambers

**Affiliations:** 1Division of Population Science, Department of Medical Oncology, Thomas Jefferson University, Philadelphia, PA 19107, USAmelissa.dicarlo@jefferson.edu (M.D.); hattie.jackson@jefferson.edu (H.J.); amanda.indictor@jefferson.edu (A.I.); 2Division of Biostatistics, Department of Pharmacology, Physiology, and Cancer Biology, Thomas Jefferson University, Philadelphia, PA 19107, USA; ayako.shimada@jefferson.edu; 3Department of Family and Community Medicine, Thomas Jefferson University, Philadelphia, PA 19107, USA; kaitlyn.davis@jefferson.edu (K.D.); william.leach@jefferson.edu (W.T.L.); christopher.chambers@jefferson.edu (C.V.C.)

**Keywords:** multi-cancer early detection tests, primary care, cancer screening

## Abstract

Multi-cancer early detection (MCED) tests are being developed, but little is known about patient receptivity to their use for cancer screening. The current study assessed patient interest in such testing. Our team conducted a prospective, observational study among primary care patients in a large, urban health system. They were asked to complete a telephone survey that briefly described a new blood test in development to identify multiple types of cancer, but was not currently recommended or covered by insurance. The survey included items to assess respondent background characteristics, perceptions about MCED testing, and interest in having such an MCED test. We also used multivariable analyses to identify factors associated with patient interest in test use. In 2023, we surveyed 159 (32%) of 500 identified patients. Among respondents, 125 (79%) reported a high level of interest in having an MCED test. Interest was not associated with personal background characteristics, but was positively associated with the following expectations: testing would be recommended for cancer screening, be convenient, and be effective in finding early-stage disease (OR = 11.70, 95% CI: 4.02, 34.04, *p* < 0.001). Research is needed to assess patient interest and actual uptake when detailed information on testing is presented in routine care.

## 1. Introduction

Multi-cancer detection (MCED) tests that evaluate cell-free DNA (cfDNA) or other circulating biomarkers are being developed for use in cancer screening [1,2]. Clinical trials have been initiated and are being planned to assess the potential benefits (e.g., reduced cancer-specific mortality) and harms (e.g., unnecessary procedures and over-diagnosis) of MCED testing [3]. As part of the Cancer MoonshotSM initiative, the National Cancer Institute (NCI) aims to conduct a large clinical trial (Vanguard) to determine the safety and efficacy of selected MCED blood tests for detecting cancer and preventing cancer-related deaths [4].

Data from clinical trials on the safety and efficacy of MCED testing are needed to inform approval by the Food and Drug Administration (FDA) and guide decision making about utilization. There are currently only four cancers for which there are standard-of-care screening tests: breast, colorectal, lung, and cervical [5,6,7,8]. If shown to be safe and effective, MCED tests could facilitate the early detection of cancers for which site-specific screening tests are not currently available and cancers that are missed by existing screening modalities [9,10]. Assuming that findings from ongoing and planned trials support MCED test use in cancer screening, it is likely that there will be a strong push for health systems to implement such testing in clinical care [11]. Unfortunately, there is a paucity of information on perceptions related to MCED testing among patients in primary care practice settings, where most standard-of-care cancer screening takes place.

A recent scoping review found only one published study in which patients were asked to indicate their receptivity to having a multi-cancer test for DNA markers [12,13]. Gelhorn et al. reported results of a web-based survey of older adults in the United States designed to assess their receptivity to MCED testing and desired characteristics of such testing [14]. Importantly, only the latter study provided survey respondents with information about blood-based MCED testing. Findings from both investigations suggest that older adults are interested in MCED testing.

A systematic review of blood-based MCED testing in populations highlights the need for research that aims to address existing gaps in knowledge related to patient receptivity to MCED testing [15]. The current study was conducted to assess the receptivity of older adult primary care patients to MCED blood testing and to identify factors that are likely to influence test uptake.

## 2. Materials and Methods

### 2.1. Study Design 

Our research team conducted a prospective, observational study from January 2023 to June 2023 among patients in three primary care practices (one internal medicine and two family medicine primary care practices) of Jefferson Health, a large healthcare system in Philadelphia. The study was approved by the Jefferson Institutional Review Board (IRB #21C.806) and all participants provided informed consent.

### 2.2. Participants 

Eligible participants included primary care patients identified via electronic medical record (EMR) data, who were between 50 and 80 years of age, had no prior diagnosis of cancer, and had a scheduled primary care office visit within two to three weeks at the time of EMR ascertainment.

### 2.3. Procedures

Patients were identified through weekly queries of the health system electronic medical record (EMR) system. For each cohort, the research team provided a list of patients to their primary care provider, who confirmed eligibility for contact. Study research coordinators attempted telephone contact with patients who were retained in the sample. During the call, the research coordinator introduced the study, verified eligibility, obtained verbal consent, and administered a survey questionnaire. The survey was designed to collect information on participant sociodemographic background characteristics, perceptions about MCED testing, and interest in having an MCED test.

At the beginning of the survey, the research coordinator used the following script to describe MCED testing to the patient: “Research is underway to develop a new blood test to detect different types of cancer. This type of test is called a multi-cancer early detection (MCED) blood test. In MCED testing, a blood sample is drawn and analyzed in a laboratory. A positive (abnormal) MCED test result is followed by a full-body CT scan to find out if and where there is a cancer. MCED testing can be done along with “standard of care” screening (i.e., mammography for breast cancer, Pap testing for cervical cancer, colonoscopy for colorectal cancer, low-dose CT scan for lung cancer). Standard of care screening is recommended and is normally covered by insurance. MCED testing is still being evaluated. As a result, it is not currently recommended as standard of care screening and is not covered by insurance.” Furthermore, the research coordinator explained that MCED testing is being offered in some settings. Subsequently, study participants were asked to indicate their level of interest in having an MCED test on a scale of 0–10 (0 = Extremely Low Interest and 10 = Extremely High Interest). An open-ended question was included to allow respondents to indicate the top three reasons for their response (i.e., decision factors).

The survey also included 14 items intended to assess respondent perceptions and attitudes towards cancer and having an MCED test. Items were based on Preventive Health Model (PHM) statements, which we have classified into cognitive (i.e., perceived salience, convenience, and response efficacy), affective (i.e., fears, worries and concerns plus perceived risk and susceptibility), and social (provider support and influence) constructs [16]. Note: the original PHM “coherence” construct is referred to here as “convenience”, as it applies to how easy it would be for patients to have an MCED blood test. Participants were asked to respond to each item using a 5-point Likert-type response set (1 = Strongly Disagree and 5 = Strongly Agree).

Study participants received $25 remuneration via ClinCard for survey completion.

### 2.4. Data Analysis 

The study’s primary aim was to assess patient interest in having an MCED test. Initially, the distribution of participant responses to the relevant item included on the survey was evaluated to determine if the measure should be considered as a continuous or categorical variable.

Exploratory factor analyses (EFA) were conducted with the 14 PHM survey items to investigate underlying constructs (or factors) of patient perceptions related to MCED testing. Factor extraction was performed using the iterated principal factors method, and various rotations were explored to enhance interpretability [17]. Items were removed iteratively to address concerns such as low or complex loadings or theoretical concerns.

Multivariable logistic regression analyses were performed to evaluate the association of participant sociodemographic background characteristics and perceptions related to having an MCED test in the future.

MCED testing decision factors reported by respondents were summarized descriptively. We categorized the reported decision factors into PHM themes (i.e., cognitive and affective) and subthemes (i.e., salience and convenience; efficacy and effectiveness; social support and influence; fears, worries, and concerns; and perceived risk and susceptibility). Distributions of these decision factors were assessed for all participants and for the participant subgroups of those who were and were not interested in testing.

## 3. Results

Figure 1 shows that between January and June 2023, we identified 1260 unique patients through the EMR who were potentially eligible for the study, and we randomly selected 500 for outreach contact. The research team reached 309 (62%) patients in this denominator. We were able to consent and complete a survey with 159 (32%) patients selected for outreach contact.

The background characteristics of survey respondents and non-respondents are presented in Table 1. The distribution of sociodemographic background characteristics among respondents was as follows: age (mean = 64.4 years), female (67%), white (65%), married (55%), and > high school education (62%). Almost all respondents were covered by private insurance, Medicare, or Medicaid (99%). The majority of respondents had never smoked (53%) or had formerly smoked (36%), while 11% reported that they currently smoke. Inspection of background differences indicates that women were more likely than men to be survey respondents.

Table 2 displays participant perceptions of and interest in having an MCED test. The mean scores for single PHM items that reflected favorable beliefs about the salience, convenience, and efficacy of MCED testing (Q1–Q5 and Q7), that early stage cancer can be cured (Q6, and those indicating perceived provider support and influence for MCED testing (Q8–Q9) were all >4.5 of 5. Mean scores for PHM items that reflected respondent fears, worries, and concerns related to testing (Q10, Q11, Q13, and A14) were <2.8 out of 5. Concern about insurance coverage for MCED testing (Q12), however, was 3.5 out of 5.

On average, we found that interest in MCED testing was very high among respondents (8.4 out of 10). In multivariable data analyses, we dichotomized responses as ‘Low to Moderate’ (0–6) or ‘High’ (7–10). When dichotomized, we determined that 34 (21%) respondents reported low/moderate interest in having an MCED test and 125 (78.6%) had high interest in testing.

Exploratory factor analyses (EFA) resulted in a one-factor structure that reflected personal beliefs related to having MCED test we refer to as “Salience, Convenience, and Efficacy (Salience—Q1: MCED test makes sense and Q3: MCED test not important (reverse coded); Convenience—Q4: MCED test convenient; and Efficacy—Q5: MCED test can help protect health and Q7: MCED test can find early-stage cancer). The mean score for this subscale, which was calculated by averaging constituent items, displayed high reliability (Cronbach’s alpha = 0.77) and was very high (4.7).

Table 3 summarizes findings from multivariable logistic regression that models a high level of interest in having an MCED test. In terms of participant background characteristics, we found that interest in MCED testing did not vary significantly by primary care practice (*p* = 0.230), age (*p* = 0.930), sex (*p* = 0.716), race/ethnicity (*p* = 0.931), education level (*p* = 0.404), marital status (*p* = 0.292), insurance coverage (*p* = 0.300), or smoking status (*p* = 0.277). We determined that the perceived test salience, convenience, and efficacy subscale score was strongly and positively associated with reported high interest in MCED testing (*p* < 0.001). We also looked at multivariable logistic regression models that included the perceived test salience, convenience, and efficacy subscale score and the remaining 14 PHM single items, as well as participant background characteristics. However, none of the single items was significantly associated with a high interest in MCED testing (results not shown). Furthermore, we found no interactions involving patient background characteristics and the perceived test salience, convenience, and efficacy.

Finally, data displayed in Table 4 show decision factor themes and subthemes related to participant interest in MCED testing. Overall, we observed that 73% of reported decision factors were cognitive in nature, and this finding was comparable among participants with a high and low/moderate level of interest in testing. Examples of reported cognitive factors include, “I want to know the status of my health,” “Detecting cancer early is important,” and “Testing would be convenient.” In terms of reported affective factors, some participants noted, “It would be scary to be screened for cancer,” “I am worried about insurance coverage,” and “I am concerned about the accuracy of the test.” Further review showed that the perceived salience and convenience of having an MCED test was the most common cognitive factor for all participants and for the subgroup reporting a high interest in testing, while in the subgroup of participants with a low/moderate interest in testing, no subtheme was predominant. In terms of affective factors, perceived risk and susceptibility was the most common subtheme (59%). We also found that among participants with a high level of interest in testing, perceived risk and susceptibility was the most predominant affective subtheme (67%); while among those with low/moderate interest, fears, worries, and concerns was the most frequently reported affective subtheme (75%).

## 4. Discussion

The current study is the first report on primary care patient receptivity to having an MCED blood test for cancer screening. This investigation is also novel in that it provides new information on primary care patient perceptions about MCED testing and factors associated with interest in such testing. Elsewhere, published studies have focused on the knowledge, attitudes, and beliefs of diagnosed cancer patients about genomic testing and personalized medicine [18,19,20].

In this study, we determined that 79% of study participants reported that they were interested in MCED testing. This finding is consistent with results reported by Gelhorn et al. in a web-based survey of US adults who were 50 to 80 years of age who viewed a video presentation on MCED testing [13,14]. In that study, 72% of respondents indicated that they would rather have an MCED test for cancer screening than have no cancer screening. Importantly, MCED testing in the future is likely to be offered in concert with recommended standard-of-care cancer screening in primary care settings, rather than as an alternative. Research on patient receptivity to MCED testing in this context is important, as primary care is the setting in which most cancer screening will most likely take place. The current study helps to enhance our understanding not only of primary care patient receptivity to MCED testing, but also of patient perceptions that might influence test uptake.

We found that patient interest in having an MCED test did not vary by sociodemographic background. We also determined that patient interest in MCED testing was not significantly associated with certain health beliefs related to screening (i.e., belief that one’s risk for cancer is low, belief that testing might detect cancer, belief that early-stage cancer can be cured, belief that testing could find early-stage cancer, and the belief that their physician would be likely to recommend MCED testing). It should be noted that these beliefs were strongly held by most study participants, and the lack of variability in these measures, along with the strong interest in MCED testing among participants, may have limited our capacity to discern impact.

Interestingly, patient interest in MCED testing also did not vary significantly according to other patient beliefs and attitudes, including patient fears, worries, and concerns related to testing. This finding may be attributed to the description of MCED testing offered to study participants at the beginning of the survey, which explained that screening involved a simple blood test, a procedure that may not have raised anxiety. In addition, respondents may have viewed MCED testing as a positive alternative to standard-of-care screening tests.

Study participants who believed research that is underway is likely to show that MCED testing is a salient, convenient, and effective screening modality were much more likely to express interest in having this type of test than those who did not hold this view. It is reasonable to assume that patients who held the former point of view would be more receptive to having an MCED test for cancer screening than those who did not. It is interesting to note that the description of testing included in the survey mentioned that testing was not currently covered by insurance. Furthermore, the description did not provide information related to estimated costs related to initial testing and the follow-up of an abnormal test result. The absence of information on these and other concerns related to having an MCED test in the information provided to patients may have served to reinforce positive perceptions of testing.

The analysis of open-ended survey responses showed that patients who had a high level of interest in MCED testing were motivated by positive cognitive factors; while those who had low/moderate interest in MCED testing were influenced by affective factors, such as concerns about test accuracy and worry about test cost. It appears that such views gave some patients pause as they considered the prospect of MCED testing. The range of factors that are likely to influence patient receptivity to MCED testing should be explored in future studies, when more complete information about the test being offered (e.g., test performance characteristics, diagnostic follow-up procedures, and insurance coverage) is available.

Selby, Elwyn, and Volk have highlighted the need to provide complete and balanced information to patients when MCED testing is offered. They also called for research to identify patient perceptions about MCED testing and encourage the use of shared decision making about MCED test use when such tests are offered in clinical care [21]. A recent systematic review noted that when shared decision making tools are used to engage patients in considering whether or not to have cancer screening, patient knowledge increases, decisional conflict decreases, and intention to get screened increases; and these effects are more pronounced among patients from disadvantaged populations than from more advantaged populations [22]. Research is needed to explore the impact of shared decision making in the context of MCED testing for cancer screening.

There are several limitations to the current study. First, the survey response rate was low and the study was conducted with patients in only three practices of one health system. Also, most respondents did report having insurance coverage, which does not represent a diverse population. For these reasons, findings may have limited generalizability. In addition, limited information on the pros and cons of MCED testing was provided. Furthermore, respondents were asked to share their views about having an MCED test at the time when this new approach to cancer screening had been shown to be safe and effective. It is reasonable to assume that patient perceptions related to MCED testing would have been influenced by the presentation of information about attributes such as the cost of testing, the nature of diagnostic follow-up of abnormal test results, guidelines related to test use in concert with other currently recommended screening modalities, and the potential for over-diagnosis. Furthermore, patient level of interest in MCED testing might also have been affected by related patient worries and concerns.

To address the generalizability concerns noted above, research should be conducted on interest in MCED testing among larger numbers of primary care patients in a wider range of primary care practices and health systems. It would also be beneficial if such studies included the provision of more detailed information to patients about the performance characteristics and potential benefits and harms of MCED testing.

## 5. Conclusions

MCED is an emerging technology that may be used in concert with standard-of-care screening modalities to detect a range of early stage cancers for which early diagnosis and treatment has been shown to reduce mortality. The use of MCED testing may also help to reduce mortality of cancers for which there are no currently recommended screening tests. Findings from the current study suggest that primary care patients are interested in MCED testing, and patient receptivity is conditioned by perceptions related to test convenience and efficacy. The conduct of clinical trials on MCED testing that are being planned or are currently underway should be complemented by research to identify factors that are likely to influence patient uptake of testing and adherence to recommended follow-up, and studies of how to engage patients in shared decision making about MCED testing. Health systems have a unique opportunity to support research in these areas and implement strategies that benefit providers and patients [23,24].

MCED test implementation in routine clinical practice is limited at the present time, pending the completion of current and planned clinical trials and determination of test use on clinical outcomes. Nonetheless, there is a strong push to make MCED testing more widely available [25,26,27,28]. Findings from the current study suggest that when patients are provided information about MCED testing that focuses on the describing the logistics of having a “simple” blood test for cancer screening, patient interest in having MCED testing is likely to be very high. Before making MCED testing widely available to primary care patients, however, health systems and health care providers should consider not only the results of clinical trials, but also how to address the need to provide patients with more complete information about the pros and cons of MCED testing, including details related to test use in concert with other types of recommended cancer screening tests, procedures required for following up abnormal MCED test findings, and insurance coverage and costs associated with testing and follow-up [29]. Attention should also be paid to addressing provider concerns related to the time required for patient education and decision support, along with the management of false positive and false negative findings. Finally, we should determine how to ensure equity related MCED testing in diverse patient populations [30].

## Figures and Tables

**Figure 1 jpm-13-01613-f001:**
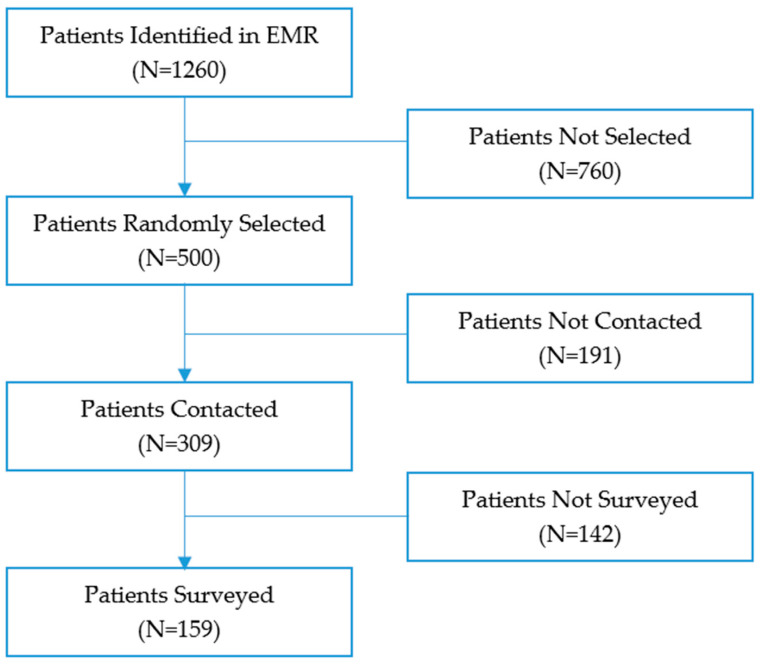
Study Design.

**Table 1 jpm-13-01613-t001:** Characteristics of Survey Respondents (N = 159) and Non-respondents (N = 341).

	Respondents		Non-Respondents	
Age (EMR) (yrs), mean (sd)	64.4	7.9	64.7	8.1
Age (EMR) (yrs), n (%)				
50–59	45	28.3%	107	31.4%
60–69	65	40.9%	129	37.8%
70–80	49	30.8%	105	30.8%
Sex (EMR), n (%)				
Female	106	66.7%	179	52.5%
Male	53	33.3%	162	47.5%
Sex (SURVEY), n (%)			NA	
Female	106	66.7%		
Male	53	33.3%		
Race/ethnicity (EMR), n (%)				
White	103	66.9%	199	60.7%
African American	42	27.3%	105	32.0%
Hispanic/Latino	7	4.5%	13	4.0%
Asian	2	1.3%	11	3.4%
Other	0	0.0%	0	0.0%
Race/ethnicity (SURVEY), n (%)			NA	
White	104	65.4%		
African American	39	24.5%		
Hispanic/Latino	11	6.9%		
Asian	2	1.3%		
Other	3	1.9%		
Marital status (SURVEY), n (%)			NA	
Never married	39	24.5%		
Separated/Divorced	22	13.8%		
Widowed	11	6.9%		
Married/Living as married	87	54.7%		
Education (SURVEY), n (%)			NA	
High school degree/GED or less	59	37.8%		
Associate’s degree or some college	23	14.7%		
College graduate and above	74	47.4%		
Insurance (SURVEY) *, n (%)			NA	
Private	104	65.8%		
Medicare	76	48.1%		
Medicaid	17	10.8%		
No insurance	1	0.6%		
Smoking status (EMR), n (%)				
Never smoker	84	53.2%	198	59.1%
Former smoker	57	36.1%	94	28.1%
Current smoker	17	10.8%	43	12.8%

* Categories not mutually exclusive (multiple insurance sources possible).

**Table 2 jpm-13-01613-t002:** Survey Results (N = 159).

	Initial Response		Range and Reliability
**Overall Score for PHM Items mean (sd)**	4.2	(0.4)	Range = 2.6–4.9. Alpha = 0.54
Single PHM items, mean (sd)			
Q1. MCED test makes sense	4.7	(0.7)	
Q2. MCED test too much time *	4.6	(1.0)	
Q3. MCED test not important *	4.5	(0.9)	
Q4. MCED test convenient	4.6	(0.8)	
Q5. MCED test can help protect health	4.7	(0.7)	
Q6. Early-stage cancer is curable	4.7	(0.7)	
Q7. MCED test can find early-stage cancer	4.8	(0.5)	
Q8. Doctor would recommend MCED test	4.5	(0.9)	
Q9. Would follow doctor’s advice about MCED test	4.8	(0.5)	
Q10. Concerned MCED test not safe or effective **	2.6	(1.1)	
Q11. Afraid of abnormal MCED test result **	2.5	(1.6)	
Q12. Concerned MCED test not covered by insurance **	3.5	(1.4)	
Q13. Believe MCED test would show I have cancer **	2.8	(1.4)	
Q14. Believe my chance of cancer is low ***	1.9	(1.2)	
Interest in MCED test, mean (sd)	8.4	(2.1)	
Interest in MCED test, n (%)			Range = 0–10, med = 10
Moderate (0–6)	34	(21.4%)	
High (7–10)	125	(78.6%)	

* Items reverse-coded when computing the overall and scale score. ** Items reverse-coded when computing the overall score. *** Items reverse-coded when computing the scale score.

**Table 3 jpm-13-01613-t003:** Predictors of interest in MCED test (N = 155).

		Interested in MCD Test				
	*N*	*n*	%	*OR*	(95% CI)	*p*
Visit practice						0.230
Abington Plaza (IM)	48	41	85.4%	1	REF	
Bensalem (FM)	54	45	83.3%	0.50	(0.14, 1.86)	0.304
JFMA/GER (FM)	53	36	67.9%	0.32	(0.08, 1.19)	0.088
Age (yrs)						0.930
50–59	43	34	79.1%	1	REF	
60–69	65	53	81.5%	0.92	(0.26, 3.17)	0.890
70–80	47	35	74.5%	0.76	(0.17, 3.36)	0.715
Sex						
Female	103	82	79.6%	1	REF	
Male	52	40	76.9%	0.83	(0.30, 2.31)	0.716
Race/ethnicity						0.931
White	103	83	80.6%	1	REF	
African American	37	27	73.0%	0.78	(0.20, 2.99)	0.716
Other	15	12	80.0%	0.85	(0.15, 4.86)	0.852
Marital status						
Never married/separated/divorced/widowed	70	54	77.1%	1	REF	
Married/living as married	85	68	80.0%	1.71	(0.63, 4.63)	0.292
Education						0.404
High school degree/GED or less	58	48	82.8%	1	REF	
Associate’s degree or some college	23	19	82.6%	0.63	(0.15, 2.73)	0.540
College graduate and above	74	55	74.3%	0.47	(0.16, 1.41)	0.179
Insurance						
Private	70	58	82.9%	1	REF	
Other (Medicare/Medicaid/no insurance)	85	64	75.3%	0.53	(0.16, 1.76)	0.300
Smoking status						0.277
Never smoker	81	61	75.3%	1	REF	
Former smoker	57	47	82.5%	2.44	(0.82, 7.27)	0.111
Current smoker	17	14	82.4%	1.20	(0.25, 5.88)	0.823
Perceptions regarding MCD testing						
Salience, convenience, and efficacy subscale (sd)	4.7	(0.5)		11.70	(4.02, 34.04)	<0.001

OR: adjusted odds ratio (model included all variables shown). CI: confidence interval.

**Table 4 jpm-13-01613-t004:** Distribution of PHM decision factor themes and subthemes by participant interest in MCED testing.

Theme	Subtheme	Total Number of Reported Decision Factors		Factors of Those with High Interest in MCED Testing		Factors of Those with Low/Moderate Interest in MCED Testing	
		*N*	(%)	*n*	(%)	*n*	(%)
TOTAL		219		176	(80%)	43	(20%)
Cognitive		156	(73%)	125	(71%)	31	(72%)
Affective		63	(27%)	51	(29%)	12	(28%)
Cognitive	Total	156		125		31	
	Salience and Convenience	112	(72%)	100	(80%)	12	(39%)
	Efficacy and Effectiveness	28	(18%)	16	(13%)	12	(39%)
	Social Support and Influence	16	(10%)	09	(07%)	07	(22%)
Affective	Total	63		51		12	
	Fears, Worries, and Concerns	26	(41%)	17	(33%)	09	(75%)
	Risk and Susceptibility	37	(59%)	34	(67%)	03	(25%)

## Data Availability

The data are available upon request.

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
