# Peer review of "Primary Care Patient Interest in Multi-Cancer Early Detection for Cancer Screening"

_jpm, 2023, doi:10.3390/jpm13111613_

Round 1

Reviewer 1 Report

Comments and Suggestions for Authors

Authors Ronald Myers and colleagues have submitted a manuscript titled “Primary Care Patient Interest in Multi-Cancer Early Detection for Cancer Screening.” They have performed patient surveys and follow-up statistical analyses to estimate the likelihood of patients subscribing for MCED tests and the factors associated with their response. This reviewer is expressing their concerns below.

Major Concerns:

i)                 There are two major concerns with this study. The first one is that as the patients are all restricted within three health practices, the findings can hardly be generalized. In the opinion of this reviewer, the choice of sample in these kinds of studies needs to be more robust and diverse in order to draw any consensus conclusions about any type of trends. The authors describe this issue in the last part of their discussion, however, that does not solve the problem of this article, that is, how much of this finding can be dependable in various socio-economic contexts.

ii)                The second concern is a bit graver. The authors do not provide enough information in the paper about what information about the MCED test was provided to the patients. They mention in the conclusion that “… limited information on the pros and cons of MCED test was provided.” This is the most confusing part of the paper, which needs to be clarified. In the discussion session, the authors mention that the patients that responded positively towards MCED test can be motivated to detect cancer “early”. As a matter of fact, this reviewer is not entirely sure of the prospect of this promise. Early cancer cells in a lot of organs (such as, ovaries) may not secrete enough circulating tumor DNA to be detected by MCED. Therefore, a positive MCED test may not always correctly correlate with the “early” occurrence of cancer. In this case, earlier detection can seem more ‘curable’ to patients driving positive results, therefore, this part cannot be left obscure.

Minor Concerns:

i)                 Standard-of-care is better hyphenated following standard practice.

ii)                Page 8, line 208 has a typo `MCD test’. Also, the references at the end of the sentence should come before the period mark.

Author Response

Reviewer 1 noted that the paper presents survey data from patients in three primary care practices, and correctly points out that the findings cannot be generalized to primary care patients across the wide range of primary care practices. We completely agree. In fact, as noted by Reviewer 1, we acknowledge this limitation in the Discussion stating, “. . . the study was conducted with patients in only three practices in one health system. As a result, findings have limited generalizability.” The current study is unique, however, in that to our knowledge it is the only report that has focused attention on the assessment of primary care patient interest in MCED test use for cancer screening. Nonetheless, we have added text to the Discussion section of the manuscript that calls for research on interest in MCED testing among larger numbers of primary care patients and in a wider range of primary care practices and health systems.
In addition, Reviewer 1 pointed out that the manuscript did not provide enough information about the description of MCED testing that was provided to patients. Again, we agree with this comment and have attempted to address this issue in Section 2.3 of the manuscript. Specifically, we have added the following text, which was used in the survey to describe MCED testing to participants, “Research is underway to develop a new blood test to detect different types of cancer. This type of test is called a multi-cancer early detection (MCED) blood test. In MCED testing, a blood sample is drawn and analyzed in a laboratory. A positive (abnormal) MCED test result is followed by a full-body CT scan to find out if and where there is a cancer. MCED testing can be done along with “standard of care” screening (i.e., mammography for breast cancer, Pap testing for cervical cancer, colonoscopy for colorectal cancer, low-dose CT scan for lung cancer). Standard of care screening is recommended and is normally covered by insurance. MCED testing is still being evaluated. As a result, it is not currently recommended as standard of care screening and is not covered by insurance.” We also agree with the sense of caution expressed by the reviewer about the
limitations of MCED testing, which is why we call for providing more detailed information about MCED testing in future research on this topic. Our view on this topic also influenced us to include text that highlights the importance of shared decision making related to the use of MCED testing in the future.
Reviewer 1 also suggested that we hyphenate “standard-of-care” following standard practice. We have made this change.
Furthermore, Reviewer 1 suggested that we should correct the use of “MCD test” on page 8 and place references before the period mark at the end of sentences. We have made those changes.

Reviewer 2 Report

Comments and Suggestions for Authors

This is a nice paper on a timely topic.   As authors rightly point out, while clinical trials to demonstrate efficacy of MCEDs are ongoing (and appear promising) there is growing need to understand how MCEDs might be perceived by patients as this would affect uptake.  RE: Table 2, understood that some questions were 'reverse coded' because of how there were asked.  As a reader, however, the scores as presented are confusing.  Suggest maybe grouping questions where lower scores denote positive responses.   Concerns I had reading the paper were addressed in the Discussion.  I agree that not having presented information to participants up-front  about the potential cost and follow-up tests following a positive screens may have skewed response.  Authors could have asked to what extent these factors might change their thoughts on test.  It is also notable that very few participants were uninsured, and population was not very diverse. 

A few grammar errors - line 259 , missing a '.'; line 262 missing 'be' safe

Author Response

Reviewer 2 commented that scores on some PHM items presented in Table 2 were confusing, as the reverse-coded tabulations were not displayed. We have addressed this matter by changing the heading for the score column from “Respondents (N=159)” to “Initial response”. We have also elaborated the text at the bottom of the table for each item designated with one or more asterisks by stating, “The initial response to this item was reverse-coded when computing the overall scale and subscale scores.”
In addition, Reviewer 2 expressed the important point that survey respondents did not reflect a diverse population, especially in terms of insurance coverage. We have added a comment on this point as a study limitation. It is worth noting, however, that 33% of survey respondents were white, a percentage that is comparable to that of primary care patients across the JH health system. Further, we believe that the distribution of other background characteristics (i.e., sex, marital status, and education) among survey respondents does reflect some level of diversity. As noted in our response to a related comment by Reviewer 1, we encourage future research in more diverse patient populations and settings.
Reviewer 2 also noted a grammatical error on line 259, “These range of factors that are likely . . .” and “a missing “. We have corrected both errors by beginning the sentence with, “Factors that are likely . . .”
Finally, Reviewer 2 observed that a word (“be”) is missing from the phrase “be safe . . .” We have corrected this error.

Round 2

Reviewer 1 Report

Comments and Suggestions for Authors

Agree to the rebuttal